# Platelets Increase the Expression of PD-L1 in Ovarian Cancer

**DOI:** 10.3390/cancers14102498

**Published:** 2022-05-19

**Authors:** Min Soon Cho, Hani Lee, Ricardo Gonzalez-Delgado, Dan Li, Tomoyuki Sasano, Wendolyn Carlos-Alcalde, Qing Ma, Jinsong Liu, Anil K. Sood, Vahid Afshar-Kharghan

**Affiliations:** 1Section of Benign Hematology, University of Texas MD Anderson Cancer Center, Houston, TX 77030, USA; hlee23@mdanderson.org (H.L.); rgonzalez9@mdanderson.org (R.G.-D.); wcarlos@mdanderson.org (W.C.-A.); vakharghan@mdanderson.org (V.A.-K.); 2Department of Hematopoietic Biology and Malignancy, University of Texas MD Anderson Cancer Center, Houston, TX 77030, USA; danli@mdanderson.org (D.L.); qma@mdanderson.org (Q.M.); 3Department of Gynecologic Oncology and Reproductive Medicine, University of Texas MD Anderson Cancer Center, Houston, TX 77030, USA; sasano106@gmail.com (T.S.); asood@mdanderson.org (A.K.S.); 4Department of Pathology, University of Texas MD Anderson Cancer Center, Houston, TX 77030, USA; jliu@mdanderson.org

**Keywords:** platelet, PD-L1, ovarian cancer, immunosuppression, NF-κB, TGFβR1, tumor microenvironment

## Abstract

**Simple Summary:**

One-third of patients with ovarian cancer have elevated platelet counts associated with a poor prognosis. We found that platelets increase the expression of immune checkpoint (PD-L1) in ovarian cancer in mice and patients. Reducing platelet counts or inhibiting platelet function reduced the expression of PD-L1 in tumors. We investigated the mechanism of platelet-induced PD-L1 and showed that platelets increase PD-L1 on cancer cells both directly (contact-dependent through NF-κB signaling) and indirectly (contact-independent via TFGβ released from platelets through TFGβR1/Smad signaling). Our results show that platelets dampen the antitumor immune response in the tumor microenvironment. Based on our preclinical results, we speculate that platelet counts might be a predictive biomarker for immunotherapy, i.e., patients with thrombocytosis respond better to anti-PD1/PDL-1 therapy. On the other hand, the use of aspirin or other antiplatelet reagents may impact the effectiveness of immunotherapy. These speculations need to be examined in clinical trials.

**Abstract:**

The interactions between platelets and cancer cells activate platelets and enhance tumor growth. Platelets increase proliferation and epithelial–mesenchymal transition in cancer cells, inhibit anoikis, enhance the extravasation of cancer cells, and protect circulating tumor cells against natural killer cells. Here, we have identified another mechanism by which platelets dampen the immune attack on cancer cells. We found that platelets can blunt the antitumor immune response by increasing the expression of inhibitory immune checkpoint (PD-L1) on ovarian cancer cells in vitro and in vivo. Platelets increased PD-L1 in cancer cells via contact-dependent (through NF-κB signaling) and contact-independent (through TFGβR1/Smad signaling) pathways. Inhibition of NF-κB or TGFβR1 signaling in ovarian cancer cells abrogated platelet-induced PD-L1 expression. Reducing platelet counts or inhibiting platelet functions reduced the expression of PD-L1 in ovarian cancer. On the other hand, an increase in platelet counts increased the expression of PD-L1 in tumor-bearing mice.

## 1. Introduction

The association between elevated platelet counts and the increase in tumor growth has been recognized in numerous clinical and experimental settings [1,2,3,4]. One-third of patients with ovarian cancer have elevated platelet counts that predict a poor prognosis [1]. Our previous studies showed that cancer cells activate platelets by secreting ADP that binds to P2Y12 on platelets. Activated platelets secrete TGFβ1, increasing the proliferation of ovarian cancer cells [2,5,6]. Reduction in platelet counts or function decreased tumor growth and increased chemosensitivity [3,4,5]. Few studies demonstrated that platelets inhibit natural killer (NK) cells and subvert T cell differentiation toward regulatory T cells (Treg) [7,8,9,10]. In the current study, we show that platelets blunt immune response to ovarian cancer by increasing the expression of the negative regulatory checkpoint proteins in tumors.

Immune checkpoints are expressed on T cells and antigen-presenting cells (APCs) and regulate T cell activation. The programmed death protein-1 (PD-1) on T cells acts as an off-switch in T cell activation. [11,12]. In the tumor microenvironment, the expression of ligand for PD-1 (PD-L1) by cancer cells or other stromal cells inhibits T cells and helps cancer cells to evade immune surveillance. The expression of PD-L1 in tumors is associated with a worse prognosis in various cancer, including ovarian cancer [13]. On the other hand, blocking PD-L1 or PD-1 has improved outcomes in many cancer patients. However, immunotherapy benefits only a minority of cancer patients and is usually not long-lasting. Various factors, including NF-κB signaling, regulate the expression of PD-L1 in cancer cells [14]. Identifying the factors that alter the expression of the immune checkpoints in tumors may have a significant clinical impact by improving the efficacy of immunotherapy [14].

We and others have studied the interaction between platelets and cancer cells. This interaction activates intracellular signaling in cancer cells, including NF-κB, Smad, and STAT1/2 pathways [10]. The current study investigated the effect of platelets on PD-L1 expression in cancer cells and the intracellular signaling pathways mediating this effect. We examined the effect of altering platelet counts or function on PD-L1 expression. We found that platelets increased the expression of PD-L1 on ovarian cancer cells both in vitro and in murine models of ovarian cancer. In addition, tumor specimens resected from patients with elevated platelet counts also expressed more PD-L1 than patients with normal platelet counts.

## 2. Materials and Methods

All the studies on mice were conducted according to the protocols approved by the institutional review board (IRB) and the Institutional Animal Care and Use Committee of the University of Texas MD Anderson Cancer Center (UT-MDACC). The de-identified tumor blocks were collected under IRB-approved protocol. The informed consent was not applicable.

### 2.1. Murine and Human Cancer Cell Lines

Murine cancer cell line, ID8 (ovarian cancer cell), and B16-F10 (melanoma cancer cells) were incubated in DMEM media supplemented with 10% heat-inactivated fetal bovine serum (FBS), 1% penicillin-streptomycin (P/S), and 0.1% insulin-transferrin-sodium selenite (ITS, only for ID8) at 37 °C in a humidified incubator infused with 20% O_2_ and 5% CO_2_. Human ovarian cancer cell lines, SKOV3, OVCAR8, HeyA8, A2780, OVCAR432, and OVCAR4, were maintained in RPMI1640, and OVCAR5 cells were in DMEM media supplemented with 10% FBS, 1% P/S at 37 °C in a humidified incubator infused with 20% O_2_ and 5% CO_2_. Murine cancer cell line, 393P (lung adenocarcinoma cell, a kind gift from Dr. Jonathan M. Kurie, Department of Thoracic H&N Medicine, MD Anderson Cancer Center), various human cancer cell lines, Daudi (human B lymphoblast, ATCC), Jurkat (human T lymphoblast, ATCC), and THP1 (human monocytic leukemia cell, ATCC) were in RPMI1640 media supplemented with 10% FBS, 1% P/S at 37 °C in a humidified incubator infused with 20% O_2_ and 5% CO_2_.

### 2.2. Murine Models of Ovarian Cancer and Drug Treatment

We used a syngeneic mouse model of ovarian cancer using ID8 murine ovarian cancer cells in C57BL/6 mice and an orthotopic mouse model of ovarian cancer using human ovarian cancer cells (A2780) in *Nu*/*Nu* mice. Female *Nu*/*Nu* and C57BL/6 mice were purchased from The Jackson Laboratory. Two million ID8 or A2780 cells were resuspended in 200 µL of Hank’s balanced salt solution and injected into the peritoneal cavity of 6–8-week-old female mice. One week after injection of cancer cells, 100 mg/kg aspirin (ASA) or 90 mg/kg ticagrelor (Tica) was administrated by daily oral gavage [5]. Mice were monitored daily for 4–5 weeks for their well-being and tumor growth until they became moribund. In some experiments, to deplete platelets, mice received monoclonal anti-GPIb-α antibody (cat# R300, Emfret ANALYTICS, Würzburg, Germany) at a dose of 0.5 μg/g via tail vein injections (i.v.) twice per week until they became moribund [1,2,15]. Tumor-bearing moribund mice were sacrificed, and their tumors were resected, weighed, fixed in 10% formalin, and kept in 70% ethanol until embedding in paraffin.

### 2.3. Platelet Isolation and Co-Incubation with Cancer Cells

Whole blood samples were obtained from the inferior vena cava of anesthetized mice (about 800 μL/mice) by a 27 gauge needle into a syringe preloaded with acid-citrate dextrose (1:9 *v*/*v* anticoagulant to blood ratio) and gently mixed with 1:1 *v*/*v* tyrodes buffer (140 mM NaCl, 2.7 mM KCl, 12 mM NaHCO_3_, 6.45 mM NaH_2_PO_4_, 5.5 mM glucose in diH_2_O lacking Mg^2+^ and Ca^2+^). Platelet-rich plasma (PRP) was isolated by centrifugation of the whole blood at 50× *g* (around 500 rpm) for 20 min at room temperature. Platelets were isolated from PRP using a gel filtration column prepared from Sepharose 4B (Sigma-Aldrich, St Louis, MO, USA) and elution buffer 1 (134 mM NaCl, 12 mM NaHCO_3_, 2.9 mM KCl, 0.34 mM Na_2_HPO_4_, 1 mM MgCl_2_, 10 mM HEPES, 5 mM glucose, 0.3 g/100 mL BSA, pH 7.4) according to the manufacturer’s protocol, and counted using a Z2 counter (Beckman Coulter, Brea, CA, USA). The ovarian cancer cells and platelets were co-incubated for 48 h at 1:25, 1:50, or 1:100 ratio (cancer cells to platelets) in a 6-well dish and harvested for Western blotting after 24 hrs. For indirect exposure, platelets were added to the upper chamber in a transwell co-culture system with 0.4 μm membranes (corning, cat#3450), and cancer cells were seeded and maintained in the lower chamber. Fixed platelets were prepared by adding 2% paraformaldehyde in tyrode’s buffer to washed platelets

### 2.4. Immunohistochemistry

Immunohistochemistry (IHC) staining for PD-L1 was performed on 4 µm-thick sections of formalin-fixed; paraffin-embedded tumor nodules using the method previously described [5]. Briefly, slides of tumor sections were deparaffinized. Antigen retrieval for PD-L1 (pH 9) was performed in a humid steamed chamber. Non-specific binding was blocked with 3% H_2_O_2_ in PBS. Slides were incubated with the primary anti-PD-L1 antibody (1:500 dilution, Cell Signaling Technology, Danvers, MA, USA) for 1 h at room temperature, washed, and then incubated with the anti-Rabbit-horseradish peroxidase (HRP)-conjugated secondary antibody for 1 h, and subsequently exposed to HRP substrate (VECTOR Laboratories, VECTASTAIN^®^ ABC-HRP Kit). The intensity of IHC staining of target proteins on the tumor tissues and the percentage of stained cells were evaluated by densitometry using ImageJ (National Institutes of Health, Bethesda, MD, USA, http://rsb.info.nih.gov/ij/index.html, accessed on 20 October 2020).

### 2.5. Small Interfering RNA Transfection

Universal negative control siRNA (SIC001), predesigned human RelA siRNA (SASI_Hs01_00171091), and human TGFβR1 siRNA (SASI_Hs01_00181142) were obtained from Sigma-Aldrich. Transfection of siRNA (10 nM) was performed using Lipofectamine RNAiMAX (Thermo Fisher Scientific, Waltham, MA, USA) following the manufacturer’s instruction. Transfected cells were co-incubated in the presence or absence of platelets and used in Western blotting.

### 2.6. Treatment of p65 Inhibitors

NF-κB inhibitors, SC-514 and JSH-23, were purchased from Sigma-Aldrich. Cancer cells were incubated with SC-514 (100 μM) and JSH-23 (50 μM) for 2 h before the addition of platelets.

### 2.7. Western Blotting Analysis

Cells were lysed in RIPA buffer (Millipore Sigma, St. Louis, MI, USA, R0278) in the presence of protease inhibitors (Roche, 11836170001). Protein lysates were quantified using BCA protein assay (Thermo Scientific, 23225), denatured in 2 × Laemmli sample buffer (Biorad, Hercules, CA, USA, 161-0737) at 95 °C for 10 min, and placed on ice for 2 min. Proteins were separated by 10% SDS-polyacrylamide gel electrophoresis (PAGE) and transferred to PVDF membranes (Millipore, IPVH00010). Membranes were blocked in 5% skim milk in TBS buffer with 0.05% tween-20 (TBS-T), incubated with primary antibody against PD-L1 (Cell Signaling, #13684), p65 (Cell Signaling, #8242), phospho-p65 (Cell Signaling, #3033), phospho-STAT1(Cell Signaling, #9167), STAT1(Cell Signaling, #14994), phospho-STAT3(Cell Signaling, #9145), STAT3(Cell Signaling, #12640), Smad2/3 (Cell Signaling, #3102), phospho-Smad2/3(Cell Signaling, #8828), TGFβR1 (Santa Cruz, sc-518018), or GAPDH (Santa Cruz, Dallas, TX, USA, sc-32233) at 4 °C overnight, washed in TBS-T buffer, and incubated with HRP-conjugated secondary antibody (G.E. Healthcare, Chicago, IL, USA, NA9340) at room temperature for 1 h. Protein bands were visualized using enhanced chemiluminescence (ECL) reaction. The density of protein bands was quantified using Image J software. The density of actin bands was used to normalize the data for protein loading.

### 2.8. Statistics

All statistical analysis was performed using GraphPad Prism 9 Software. A two-tailed Student *t*-test was used to determine the statistical significance of the comparisons. Data throughout the manuscript were presented as mean ± SEM. The differences with *p* < 0.05 were considered to be statistically significant.

## 3. Results

### 3.1. Platelets Increase the Expression of PD-L1 in Ovarian Cancer In Vivo

We examined the effect of platelets on the expression of PD-L1 in murine models of ovarian cancer. We manipulated the number or function of platelets in tumor-bearing mice and examined its effect on the expression of PD-L1 in resected tumor nodules. We used both the syngeneic mouse model of ovarian cancer and the xenograft generated by human ovarian cancer cells in *Nu*/*Nu* mice. The number of platelets in mice was altered by either depleting platelets using an antiplatelet antibody (APA, 0.5 μg/g twice per week intravenous injection) or by increasing platelet counts using a thrombopoietin receptor agonist (TPO-RA, 10 μg/kg weekly subcutaneous injections) treatment. The function of platelets was manipulated by using aspirin (ASA, 100 mg/kg daily gavage) or ticagrelor (Tica, 90 mg/kg daily gavage) (Figure 1). In orthotopic *Nu*/*Nu* xenograft-bearing mice, reduction in platelet counts and inhibition of platelet activation reduced the expression of PD-L1 in the tumors (Figure 1a). The average percentage of PD-L1(+) cells in the tumors was as follows: Control = 15.59 ± 2.0%, APA = 8.37 ± 1.9%, ASA = 7.07 ± 1.4% and Tica = 9.43 ± 2.2% (Figure 1b).

In immunocompetent C57/B6 syngeneic tumor-bearing mice, platelet depletion by APA treatment reduced PD-L1 expression in tumors compared to controls (Figure 2a). The average percentage of PD-L1(+) cells in the syngeneic tumors was as follows: control = 4.5 ± 1.32% and APA = 2.5 ± 0.4% (Figure 2b). On the other hand, an increase in platelet counts in tumor-bearing mice using TPO-RA increased the expression of PD-L1 in tumors (Figure 2a,b) as compared to controls (TPO-RA = 17.2 ± 2.6%).

### 3.2. Platelets Enhance PD-L1 Expression in Ovarian Cancer Cells In Vitro

Next, we examined the effect of platelets on the expression of PD-L1 in ovarian cancer cells in vitro. The ID8 cells were co-incubated with gel-filtered mouse platelets in a ratio of 1:25, 1:50, or 1:100 (the ratio of cancer cells to platelets) for 48 h. Cancer cells alone without platelets served as a control. After several washes, cancer cells were lysed, and cell lysate was examined for the presence of PD-L1 protein using Western blotting. Co-incubation with platelets increased PD-L1 in ovarian cancer cells in a dose-dependent manner (Figure 2c).

### 3.3. Platelets-Induced NF-κB and TGFβR1/Smad Signaling Increase the Expression of PD-L1 in Human Ovarian Cancer Cells

We examined the effect of platelets on the expression of PD-L1 on several ovarian cancer cell lines (SKOV3, OVCAR8, HeyA8, A2780, OVCAR432, and OVCAR4). Platelets increased PD-L1 expression on all of these cell lines (Figure 3a). We examined the mechanism of platelet-mediated increase in PD-L1 expression in OVCAR8 and SKOV3 cells. Several transcription factors, such as STAT1/3, HIF-1α, NF-κB, AP1, and MYC, and various cytokines, including IFN-γ, and TGF-β, have been shown to regulate the expression of the PD-L1 gene in cancer cells [16,17,18]. Platelets activate NF-κB, TGFβR1/Smad, and JAK/STAT signaling in cancer cells [10], which led us to investigate their role in platelet-induced PD-L1 expression in ovarian cancer. The direct binding of platelets to cancer cells activates NF-κB signaling. The indirect effect of platelets is mediated by platelet release. TGF-β1 released from platelets activates TGFβR1/Smad signaling in cancer cells [2,6,10]. Both direct and indirect exposure to platelets increased PD-L1 expression in cancer cells (Figure 3b). The direct exposure to platelets activated NF-κB signaling in cancer cells, as seen by an increase in phosphorylation of p-65 (Figure 3b,c). Indirect exposure of cancer cells to platelets through a transwell membrane activated Smad2/3 phosphorylation (Figure 3b). We noted that phosphorylated p65 (phospho-p65) in ovarian cancer cells, OVCAR8 and SKOV3, increased dose-dependent after platelet exposure (Figure 3c). To further examine the role of NF-κB signaling in platelet-enhanced PD-L1 expression, we reduced p65 (RelA) in OVCAR8 and SKOV3 ovarian cancer cells using small interfering RNA (siRelA). P65 reduction in cancer cells reduced the impact of platelets on PD-L1 expression (Figure 3d). Additionally, NF-κB inhibitors (SCI 514 or JSH-23) diminished platelet-induced PD-L1 expression in ovarian cancer cells (Figure 3e,f). Next, we examined the role of TGF-β/Smad signaling in platelet-induced PD-L1 expression. Reduction in TGFβR1 using TGFβR1 siRNA in OVCAR8 diminished the impact of platelets on PD-L1 expression (Figure 3g).

While NF-κB and TGFβR1/Smad signaling played a role in platelet-induced PD-L1 expression, we could not find any significant change in phosphorylation of STAT-1 or STAT-3 in platelet-exposed cancer cells (Appendix A). Our results showed that platelets increased PD-L1 expression in ovarian cancer cells, partially through direct binding and activation of NF-κB signaling and partially through secreted TGFβ1 activating TGFβR1/Smad signaling.

Next, we examined whether the impact of platelets on PD-L1 expression is limited to ovarian cancer or affects other cancer cells. Co-incubation of platelets with B16-F10 (murine melanoma), 393P (murine lung adenocarcinoma), Daudi (human B lymphoblasts), Jurkat (human T lymphoblasts), and THP-1 (human monocytic cells) increased PD-L1 expression on these cells (Figure 3h,i).

### 3.4. Elevated Platelet Counts Increase the Expression of PD-L1 in Human Ovarian Cancer Tissues

We examined the effect of platelet counts on PD-L1 expression in tumor specimens resected from patients with ovarian cancer. We compared the expression of PD-L1 in tumor specimens of 10 patients with thrombocytosis (>450,000 platelets/μL) to those with normal platelet counts (<450,000 platelets/μL) using immunohistochemistry. We found that tumors from ovarian cancer patients with thrombocytosis showed a higher expression of PD-L1 than those with normal platelet counts (Figure 4). The average percentage of PD-L1(+) cells in tumors from patients with a high platelet count was 13.45 ± 2.25% and for patients with normal platelet counts, was 7.66 ± 1.8% (*p* < 0.0001, two-tailed *t*-test) [1,3,5,19].

### 3.5. Elevated Platelet Counts Enhance the Efficacy of Anti-PD-L1 Immunotherapy

We studied the effect of platelet counts on the antitumor effect of an anti-PD-L1 antibody in the syngeneic murine model of ovarian cancer. An anti-mouse PD-L1 antibody (200 μg/mouse, clone B7-H1, BioXCell) was injected through the intraperitoneal cavity of tumor-bearing mice on days 2 and 4 after the injection of murine ovarian cancer cells. Platelet counts were reduced using depletion by an antiplatelet antibody or increased using TPO-RA (Appendix A). The experimental protocols for immunotherapy in tumor-bearing mice with altered platelet counts are shown in Figure 5a,b.

The antiplatelet antibody (APA, anti-GP1b blocking antibody) at a dose of 0.5 μg/g was injected intravenously three days before the injection of cancer cells. The platelet counts dropped immediately after APA injection and remained low for 3–5 days as quantified by complete blood counts (Appendix A). We injected APA twice a week for 4–6 weeks to maintain a consistent 50% reduction in platelet counts until tumor-bearing mice became moribund and were sacrificed.

We and others have shown that platelet depletion reduces the growth of the ovarian tumor [1,3,5]. A single injection of the anti-PD-L1 antibody decreased tumor growth in the murine model of ovarian cancer [19]. The current study showed that platelet depletion alone (APA-treated mice) or immunotherapy alone reduced tumor growth, but adding them together (APA- and αPD-L1-treated mice) did not have an additional benefit. In fact, declining platelet counts decreased the therapeutic effect of anti-PD-L1. The anti-PD-L1 antibody did not have any antitumor activity in platelet-depleted tumor-bearing mice (Figure 5a, last two columns). The average total weight of tumor nodules in control (0.39 ± 0.03 g), anti-PD-L1-treated (0.26 ± 0.044 g), APA-treated (0.24 ± 0.032 g), and anti-PD-L1- and APA-treated (0.26 ± 0.021 g) mice, the standard deviations and *p*-values are shown in Figure 5a (*n* = 10 mice/group, *p*-values are calculated using two-tailed Student *t*-test).

The experimental protocol for combining TPO-RA and immunotherapy is shown in Figure 5b. TPO-RA was injected subcutaneously (s.c.) at a dose of 10 μg/kg weekly starting one week before cancer cell injection and continued for 4–6 weeks until tumor-bearing mice became moribund. TPO-RA resulted in a 30–40% increase in platelet counts, lasting for about 4–5 days. (Appendix A). An increase in platelet counts increased the tumor size (Figure 5b), consistent with our previous data [5]. Interestingly, anti-PD-L1 showed a more potent antitumor activity in TPO-RA-treated mice than in control mice with normal platelet counts (Figure 5b). The average total weight of tumor nodules in control (0.21 ± 0.008 g), anti-PD-L1-treated (0.18 ± 0.014 g), TPO-RA-treated (0.29 ± 0.023 g), and anti-PD-L1- and TPO-RA-treated (0.15 ± 0.016 g) mice, the standard deviations and *p*-values are shown in Figure 5b (*n* = 6 mice/group, *p*-values are calculated using two-tailed Student *t*-test). The data indicated that higher platelet counts increased the efficacy of immunotherapy in tumor-bearing mice.

### 3.6. Antiplatelet Drugs Reduce the Efficacy of Immunotherapy

We showed that antiplatelet reagents reduced the expression of PD-L1 in ovarian cancer. We examined whether aspirin or ticagrelor affects the efficacy of immunotherapy with an anti-PD-L1 antibody. Aspirin (100 mg/kg) or ticagrelor (90 mg/kg) was delivered by daily oral gavage to mice starting one week after injection of ovarian cancer cells and continued thought out the length of experiments (4–6 weeks). Aspirin did not reduce tumor growth in vivo but reduced the antitumor effect of the anti-PD-L1 antibody (Figure 5c). The average total weight of tumor nodules in control (0.24 ± 0.026 g), anti-PD-L1-treated (0.17 ± 0.03 g), aspirin-treated (0.23 ± 0.04 g), and anti-PD-L1 and aspirin-treated (0.19± 0.031 g) mice, the standard deviations and *p*-values are shown in Figure 5c (*n* = 9 mice/group, *p*-values are calculated using a two-tailed Student *t*-test). Ticagrelor reduced tumor growth but abolished any therapeutic benefit of anti-PD-L1 in tumor-bearing mice (Figure 5d). The average total weight of tumor nodules in control (0.23 ± 0.023 g), anti-PD-L1-treated (0.09 ± 0.025 g), ticagrelor-treated (0.16 ± 0.021 g), and anti-PD-L1- and ticagrelor-treated (0.22 ± 0.014 g) mice, the standard deviations and *p*-values are shown in Figure 5d (*n* = 5 mice/group, *p*-values are calculated using a two-tailed Student *t*-test).

## 4. Discussion

The adaptive immune response to malignant tumors limits cancer progression and may eliminate malignant cells. The activity of the adaptive immune system is tightly regulated to prevent recognition of self and constraint the extent of the response to non-self. Under physiologic conditions, inhibition of T cells prevents an overreaction of the immune system and excessive tissue injury, but inhibition of T cell activity inside the tumor microenvironment prevents antitumor immune response and promotes tumor growth. Among the regulatory checkpoints, the interaction between PD-L1 on the cell surface and PD-1 on T cells is a main inhibitory step in limiting T cell response. Several studies showed that the expression of PD-L1 by cancer cells suppresses the antitumor immune system [20,21]. As a result, antibodies blocking PD-L1 and PD-1 have become effective therapeutics in oncology.

A higher expression of PD-L1 by cancer cells may be the harbinger of a more robust response to immunotherapy [22,23,24]. However, this correlation was not uniformly present in all tumor types, and patients with little or no expression of PD-L1 may still respond to anti-PD-L1 and anti-PD-1 antibodies [25]. The expression level of immune checkpoints on cancer cells is regulated by various factors both intrinsic to cancer cells (such as genetic instability, the profile of non-coding RNAs, and post-translational modifications of PD-L1) and extrinsic to cancer cells (such as cytokine profile, inflammation, and hypoxia in the tumor microenvironment) [26]. In this study, we showed that platelets increase the expression of PD-L1 on ovarian cancer cells in vitro, in tumor-bearing mice, and in tumor specimens from ovarian cancer patients.

Previously, the immunoregulatory role of platelets in cancer has been mainly attributed to countering the direct cytotoxicity of immune cells on circulating tumor cells [27]. Platelet-coated cancer cells are not recognized by the natural killer (NK) cells. In addition to interrupting immune surveillance by NK cells [28], platelets also directly downregulate the antitumor activity of NK cells [29]. In this study, we showed another mechanism via which platelets alter the immune response in favor of malignant cells. That is by increasing the expression of inhibitory checkpoint molecules on ovarian cancer cells. Decreasing platelet counts or inhibiting platelet function reduced the expression of PD-L1 in ovarian cancer. On the other hand, increasing platelet counts using recombinant murine thrombopoietin receptor agonist (TPO-RA) increased PD-L1 expression in tumors.

An increase in the expression of PD-L1 in the tumors suppresses T cell attacks on cancer cells that may contribute to the more aggressive behavior of tumors in the presence of thrombocytosis. One might also conclude that antiplatelet reagents, such as aspirin or ticagrelor, reduce PD-L1 and enhance the antitumor immune response. However, on the other side of the token, the platelet-induced increase in the expression of immune checkpoints provides more targets for anti-PD-L1 antibodies and perhaps points to a better therapeutic response to immunotherapy. This scenario is comparable to the expression of other therapeutic targets such as the HER2 receptor on cancer cells. The expression of HER2/Neu on breast cancer cells was considered a negative prognostic biomarker due to the more aggressive behavior of the HER2(+) breast cancer than HER2(-) tumors. However, after the discovery and widespread use of HER2 blocking reagents (e.g., Trastuzumab and Pertuzumab), HER2(+) tumors are successfully treated and are even more amenable to long-term control than HER2(-) tumors.

Expression of PD-L1 on platelets of cancer patients and lack of it on normal platelets were reported [30,31,32]. The origin of cancer platelets was attributed to the transfer of PD-L1 from cancer cells to platelets [32]. It was postulated that PD-L1-expressing platelets contribute to the overall PD-L1 expression in tumors. Our flow cytometry on healthy donors’ peripheral blood did not show PD-L1 on platelets, CD4+, CD8+, or B cells. However, we detected PD-L1 expression on monocytes, natural killer (NK) cells, and dendritic cells (DC) (Appendix A). Furthermore, we could not detect PD-L1 on platelets from tumor-bearing or tumor-free mice (Appendix A). In our studies, platelets increased PD-L1 expression in those cells. Normal platelets, even when separated from ovarian cancer cells, could increase the expression of PD-L1 on cancer cells. Furthermore, recently, other investigators showed that platelets increase the expression of PD-L1 on cancer cell lines in vitro [33]. Our study showed the mechanism responsible for platelet-induced PD-L1 expression and the impact of platelet counts and function on PD-L1 expression in vivo. 

We examined the mechanism of platelet-induced expression of PD-L1 on cancer cells. The interaction between platelets and cancer cells has been known and studied for a long time; however, more recently, the molecular basis of this interaction has been understood in more detail [34,35]. Cancer cells activate platelets via different stimuli, and platelets promote proliferation, EMT, and intracellular signaling of cancer cells through various mediators. Platelets or secretory products of platelet activate NF-κB, JAK/STAT, and TGF-β/Smad pathways in cancer cells [10]. Phosphorylated P65, STAT1/3, and Smad2/3 regulate the transcription of the PD-L1 gene [16,17,18,36]. We identified an increase in phosphorylation of P65 and Smad2/3 in platelet-exposed cancer cells but did not detect a significant increase in phosphorylation of STAT1/3. Direct platelet exposure increased PD-L1 by both NF-κB and Smad2/3 signaling pathways in cancer cells, but indirect exposure to platelets increased PD-L1 mainly through TGFβR1/Smad signaling. Reduction in P65 or TGFβR1 reduced platelet-induced PD-L1 expression in cancer cells. We detected a higher expression of phosphorylated P65 and Smad2 in tumor specimens from ovarian cancer patients with thrombocytosis (Appendix A). Our results are also consistent with the reported role of NF-κB in regulating PD-L1 expression in macrophages and other immune cells [36,37].

We showed that reducing platelet counts or using antiplatelet reagents reduced the therapeutic effect of α-PD-L1 antibody in a murine model of ovarian cancer, and increasing platelet counts results in a more robust response. Based on our preclinical findings, we speculate that the use of antiplatelet reagents may reduce the efficacy of immunotherapy, and the presence of elevated platelet counts may predict a better response to immunotherapy. Both of these speculations need to be studied in future pilot clinical trials.

## 5. Conclusions

Platelets induce the expression of PD-L1 in ovarian cancer cells and dampen the antitumor immune response in the tumor microenvironment.

## Figures and Tables

**Figure 1 cancers-14-02498-f001:**
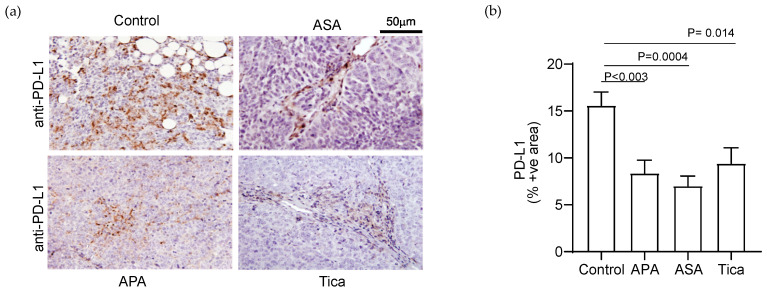
Platelets increase the expression of immune checkpoint protein, PD-L1, in the orthotopic murine model of ovarian cancer. (**a**) Representative images of PD-L1 immunostaining in tumor specimens induced by A2780 human ovarian cancer cells in *Nu*/*Nu* mice. APA: antiplatelet antibody, ASA: Aspirin, and Tica: Ticagrelor. (**b**) The relative expression of PD-L1 in tumor nodules resected from APA-, ASA-, and Tica-treated tumor-bearing mice was quantified by measuring positively stained surface area using ImageJ (National Institutes of Health, http://rsb.info.nih.gov/ij/index.html, accessed on 20 October 2020). The results are shown as the percentage of the immunostained area [PD-L1(+) area/total surface area × 100]. The statistical significance was calculated using a two-tailed *t*-test (*n* = 9 (3 tumor nodules per mouse × 3 mice)).

**Figure 2 cancers-14-02498-f002:**
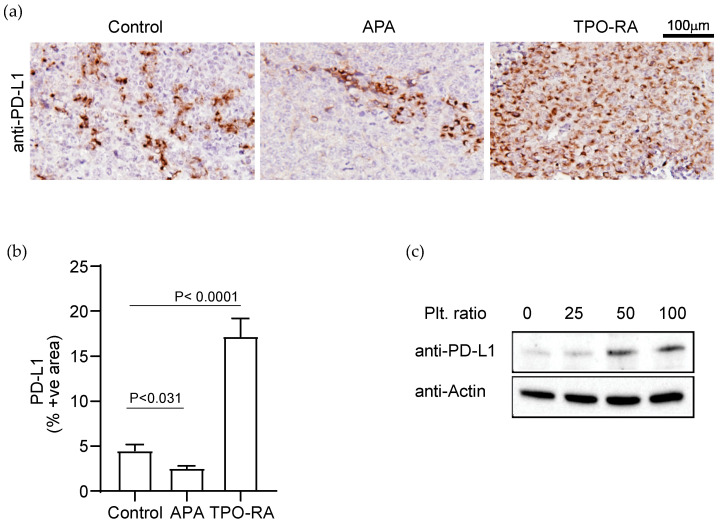
Platelets increase the expression of immune checkpoint protein, PD-L1, in the syngeneic murine model of ovarian cancer. (**a**) Representative images of PD-L1 immunostaining in tumor specimens induced by ID8 murine ovarian cancer cells in C57BL/6 mice. TPO-RA: murine thrombopoietin receptor agonist. (**b**) The relative expression of PD-L1 in tumor nodules resected from APA- and TPO-RA-treated tumor-bearing mice was quantified by measuring positively stained surface area using ImageJ. The results are shown as a percentage of the immunostained area. The statistical significance was calculated using a two-tailed *t*-test (*n* = 9 (3 tumor nodules per mouse × 3 mice)). (**c**) Western blot analysis of the expression of PD-L1 in ID8 cells at the baseline and after co-incubation with platelet at a ratio of 1:25, 1:50, and 1:100 (number of cancer cells: number of platelets). Western blot band representing β-Actin was used as a loading control. The data are representative of three independent experiments. For original blots, see Appendix A.

**Figure 3 cancers-14-02498-f003:**
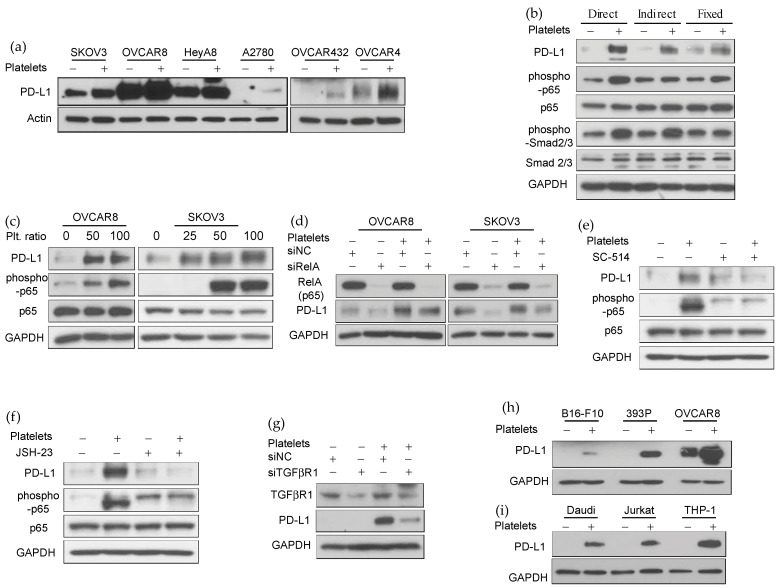
Platelets increase PD-L1 expression through NF-κB and TGFβR1/Smad signaling. (**a**) Western blot analysis of PD-L1 in various human ovarian cancer cells at the baseline and after co-incubation with platelet at a ratio of 1:100 (number of cancer cells to platelets). The protein band representing β-Actin was used as a loading control. The data are representative of three independent experiments. (**b**) Western blot analysis of whole-cell lysate prepared from cancer cells after direct incubation with washed platelets, indirect exposure to platelets separated via a membrane in a transwell chamber, or incubation with paraformaldehyde-fixed platelets. Protein bands representing GAPDH and total p65 and Smad2/3 were used as loading controls. The data are representative of three independent experiments. (**c**) Western blot analysis of whole-cell lysate from OVCAR8 and SKOV3 cells co-incubated with platelet at a ratio of 1:25, 1:50, and 1:100 (number of cancer cells to platelets). Protein bands representing GAPDH and total p65 were used as loading controls. The data are representative of three independent experiments. (**d**) Effect of p65 gene knockdown using RelA siRNA on the platelet-induced PD-L1 expression in OVCAR8 and SKOV3 human ovarian cancer cells (representative of three independent experiments). NF-κB blocking agents (**e**) 100 μM of SC-514 or (**f**) 50 μM of JSH-23 were added to ovarian cancer cells for 2 h before adding the platelets. (**g**) Effect of TGFβR1 gene knockdown on the platelet-induced PD-L1 expression in OVCAR8 human ovarian cancer cells (representative of three independent experiments). (**h**,**i**) Western blot analysis of PD-L1 in different murine and human cancer cell lines (B16-F10, 393P, OVCAR8, Daudi, Jurkat, and THP-1) baseline and after co-incubation with platelet at a ratio of 1:100 (number of cancer cells to platelets). The protein band representing GAPDH was used as a loading control (representative of three independent experiments). For original blots see Appendix A. For original densitometry results, see Appendix A.

**Figure 4 cancers-14-02498-f004:**
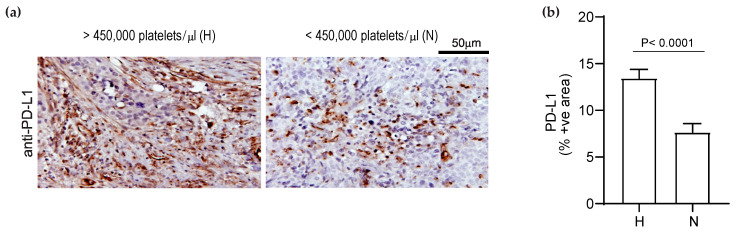
Thrombocytosis is associated with increased PD-L1 expression in human ovarian cancer tumors. (**a**) Representative immunostaining of a tumor specimen resected from a patient with thrombocytosis (platelet counts > 450,000/µL, *n* = 10 patients) and a patient with normal platelet counts (<450,000/µL, *n* = 10 patients) for PD-L1. (**b**) The comparison of expression of PD-L1 in patients with thrombocytosis (*n* = 10 patients) compared to those with normal platelet counts (*n* = 10 patients). The expression of PD-L1 was detected by immunostaining the tumor slides for PD-L1 and quantified using ImageJ software. The results are shown as the percentage of PD-L1-positive area. The statistical significance was calculated using a two-tailed *t*-test.

**Figure 5 cancers-14-02498-f005:**
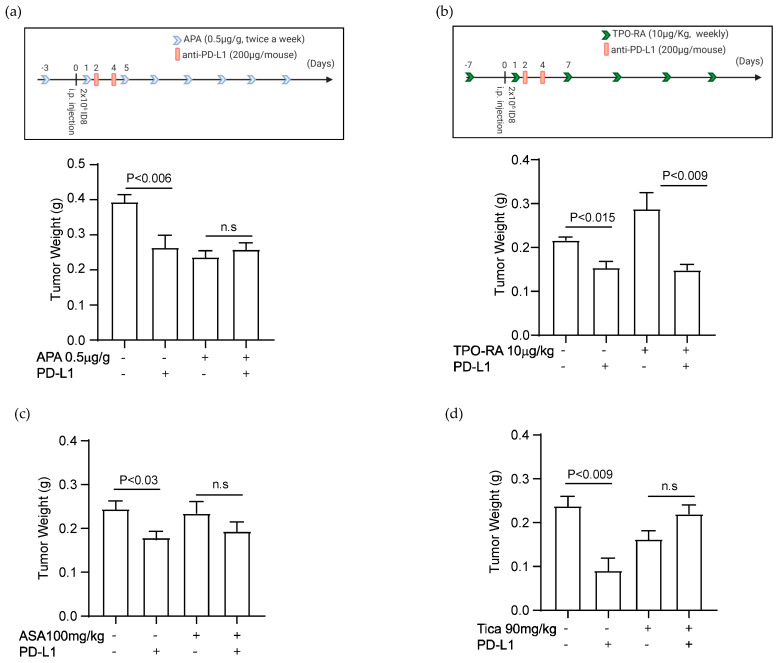
Platelet counts alter the response to immunotherapy with anti-PD-L1 antibody. The effect of antiplatelet antibody (APA), TPO-RA, Aspirin (ASA), or Ticagrelor (Tica) on the antitumor activity of anti-PD-L1 blocking antibody was examined in a syngeneic murine model of ovarian cancer. Tumors were induced by i.p. injection of ID8 cells to C57BL/6 mice. (**a**) The schematic presentation of the experimental protocol combining APA and anti-PD-L1 antibodies. The effect of combining APA and anti-PD-L1 antibody on the final total weight of tumor nodules resected from tumor-bearing mice (*n* = 10 mice/group, two-tailed *t*-test). (**b**) The schematic presentation of the experimental protocol combining TPO-RA and anti-PD-L1 antibody. The effect of combining TPO-RA and anti-PD-L1 antibody on the final total weight of tumor nodules resected from tumor-bearing mice (*n* = 10 mice/group, two-tailed *t*-test). (**c**) The effect of combining ASA (100 mg/kg daily gavage) and anti-PD-L1 antibody on the final total weight of tumor nodules resected from tumor-bearing mice (*n* = 10 mice/group, two-tailed *t*-test). (**d**) The effect of combining Tica (90 mg/kg daily gavage) and anti-PD-L1 antibody on the final total weight of tumor nodules resected from tumor-bearing mice (*n* = 10 mice/group, two-tailed *t*-test).

## Data Availability

The data supporting the conclusions of this article are available from the corresponding author upon reasonable request and with permission of the University of Texas, MD Anderson Cancer Center.

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
