# Peer review of "Platelets Increase the Expression of PD-L1 in Ovarian Cancer"

_cancers, 2022, doi:10.3390/cancers14102498_

Round 1
Reviewer 1 Report
The authors have used patient tumor specimens and used animal cell culture models to demonstrate the role played by platelets on PD-L1 expression.
The study seems concise, and authors tried to argue it to prove their hypothesis. While the role of NfkB and p65 have been extensively studied, the thought to include platelet as factor is interesting.
I have the following questions for the authors.
- Authors conclude that platelets blunt immune system? How? when platelet itself is in the fight against several diseases.
- What is the normal ration of cells: platelet in normal cells? Platelets are much smaller than most of the cells used here and the 1:100 concentration used is high or low for quantity of cells used. Can you justify.
- Did you perform any dual staining of platelets and PD-L1 to show that platelets did not come from platelets itself?. Can you show that PD-L1 is not coming from platelet itself?
- PD-L1 expression usually evades immune check points, how in this context platelets are inducing the expression of PD-L1 on cancer cells, if it exposes the PD-L1 it could be helping other immune system to sense the cancer cells. Can you justify?
- The standalone inhibition of p65 and NfKB could reduce the tumor proliferation or their survival, so irrespective of the presence of platelets this could lead to decrease in the p65 and NfkB expression. How could you confirm the role of platelets here?
- Section 2.3: Whole blood samples were obtained from the inferior vena cava of anesthetized mice (about 800 l/mice). Can you check the unit for blood?. Also check the units used throughout the manuscript.
Author Response
Authors conclude that platelets blunt immune system? How? when platelet itself is in the fight against several diseases.
Response: Immunoregulatory role of platelets has not been examined adequately. Although the anti-infection role of platelets has been proposed, several other groups and we have reported the pro-tumor effect of platelets in various cancers (PMID: 29657130). The role of platelets in protecting circulating cancer cells against NK cells has been shown before (PMID: 29308299, PMID: 30908480). Our studies showed another mechanism that platelets reduce immune attacks on cancer cells. We propose that platelets increase the expression of PD-L1 in tumors, which reduces T cell anti-tumor cytotoxicity (via PD-L1/PD1 axis).
- What is the normal ration of cells: platelet in normal cells? Platelets are much smaller than most of the cells used here and the 1:100 concentration used is high or low for quantity of cells used. Can you justify.
Response: The number of platelets in the blood is 200,000/µl. The number of platelets inside the tumor probably varies in different tumors. In our previous study, we detected 3.8~6.5 x 104 platelets / mm2 of surface area of orthotopic tumors in mice (PMID: 33821990). However, we are not aware of any data on the cancer cell: platelet ratio inside tumors, and the used ratio of 1:100 is an arbitrary one.
- Did you perform any dual staining of platelets and PD-L1 to show that platelets did not come from platelets itself? Can you show that PD-L1 is not coming from platelet itself?
Response: We performed flow cytometry on platelets and could not detect PD-L1 on platelets from normal donors (used in our studies). We added the flow cytometry data on the lack of expression of PD-L1 on normal platelets in the supplementary Figure 3. Furthermore, we could not detect PD-L1 on tumor-bearing or tumor-free mice (supplemental Figure 4)
- PD-L1 expression usually evades immune check points, how in this context platelets are inducing the expression of PD-L1 on cancer cells, if it exposes the PD-L1 it could be helping other immune system to sense the cancer cells. Can you justify?
Response: Expression of PD-L1 on cancer cells can reduce T cell cytotoxicity by binding to PD1 on T cells.
- The standalone inhibition of p65 and NfKB could reduce the tumor proliferation or their survival, so irrespective of the presence of platelets this could lead to decrease in the p65 and NfkB expression. How could you confirm the role of platelets here?
Response: We used total p65 and NfKB as the respective loading controls and normalized the results of Western-blot according to the total amount of these proteins and not the number of cells.
- Section 2.3: Whole blood samples were obtained from the inferior vena cava of anesthetized mice (about 800 l/mice). Can you check the unit for blood? Also check the units used throughout the manuscript.
Response: We thank the reviewer for checking this. We corrected all the symbols and units throughout the manuscript.
Reviewer 2 Report
In this study, the author found that the platelets increase the expression of PD-L1 in OC in mice and patient. Reducing or inhibiting platelet number or function reduced the expression of PD-L1 in the tumors. By screening multiple pathways, the author found that platelet activate PD-L1 through NF-kB pathway and TGF-b pathway. The author also found correlations between platelets counts and PD-L1 in human tumor specimen.
The manuscript is well written and easy to follow. Below are some concerns of the data that need to be addressed by the authors:
1. For the direct co-culture of tumor cells with platelets. How do the authors eliminate the PD-L1 expression from platelets in the western blot assay?
2. All the Western data need to be quantified and show statistics since some internal controls are not even.
3. In human ovarian cancer specimens, did the author see the samples with > 450,000 platelets/µ increased p-p65 or TGF-beta signaling when compared with the sample with < 450,000 platelets/µl? To better fit the idea that platelet increase cancer cell PD-L1 through activating NF-kB and TGF signaling, the authors need to check p-p65 and p-smad to see if these two pathways are activated in the samples with more platelets.
Author Response
1. For the direct co-culture of tumor cells with platelets. How do the authors eliminate the PD-L1 expression from platelets in the western blot assay?
Response: There are three points that we would like to discuss:
a. Cancer cells not directly incubated with platelets (separated by a membrane) also showed an increase in PD-L1.
b. Flow cytometry on healthy donors' platelets (used in our studies) did not show any PD-L1 expression. We added this data in supplementary figure 3 and described it in the discussion.
c. Reports on PD-L1 expression on platelets used platelets from cancer patients and did not detect PD-L1 in normal platelets (PMID: 34853305). Our studies used platelets from normal blood donors and did not detect any PD-L1 on them, as was mentioned before. Furthermore, platelets from tumor-free and tumor-bearing mice did not express PD-L1 (Supplementary Figure 4).
2. All the Western data need to be quantified and show statistics since some internal controls are not even.
Response: We performed densitometry on all Western blot bands and reported the results in an Excel file (Table S1) showing fold changes compared to control bands (all normalized to loading controls, GAPDH or Actin).
3. In human ovarian cancer specimens, did the author see the samples with > 450,000 platelets/µl increased p-p65 or TGF-beta signaling when compared with the sample with < 450,000 platelets/µl? To better fit the idea that platelet increase cancer cell PD-L1 through activating NF-kB and TGF signaling, the authors need to check p-p65 and p-smad to see if these two pathways are activated in the samples with more platelets.
Response: We had access to pathology slides from human tissue specimens and used them for IHC staining. We detected more intense staining for phospho- SMAD2 and phosphor-p65 in ovarian cancer tissues of patients with platelet counts > 450,000 platelets/µl than those with normal platelet counts. This data was added in supplementary Figure 5.
Reviewer 3 Report
In the research article entitled “Platelets increase the expression of PD-L1 in ovarian cancer.”, the authors discussed about the role of platelets in altering PDL1 expression on cancer cells. There are a few points to be noted:
The results are very compelling but before concluding anything, the authors should do an important experiment. They should use the syngeneic model to establish the tumors and treat them with Ticagrelor+ low dose aspirin to decrease PDL1 expression and TPO-RA treatment to increase PDL1 expression. Then authors should treat both models with PDL1 inhibitor. After 4 to 5 weeks of treatment, the authors should measure the tumor size emphasizing the pathological response score. They should also show the tumor number of CD8 T cells, and their proliferation using Ki67 along with a few exhaustion markers like PD1 and Tim3 using flow cytometry.
Author Response
In the research article entitled "Platelets increase the expression of PD-L1 in ovarian cancer.", the authors discussed about the role of platelets in altering PDL1 expression on cancer cells. There are a few points to be noted:
The results are very compelling but before concluding anything, the authors should do an important experiment. They should use the syngeneic model to establish the tumors and treat them with Ticagrelor+ low dose aspirin to decrease PDL1 expression and TPO-RA treatment to increase PDL1 expression. Then authors should treat both models with PDL1 inhibitor. After 4 to 5 weeks of treatment, the authors should measure the tumor size emphasizing the pathological response score. They should also show the tumor number of CD8 T cells, and their proliferation using Ki67 along with a few exhaustion markers like PD1 and Tim3 using flow cytometry.
Response: As the reviewer suggested, we performed in vivo immunotherapy experiments in tumor-bearing mice with altered platelet numbers and functions. We reported the results in Figure 5 of the revised manuscript.
Round 2
Reviewer 3 Report
The authors performed the experiment suggested and so the manuscript can be accepted for publication